# Algerian Workers’ Exposure to Mycotoxins—A Biomonitoring Study

**DOI:** 10.3390/ijerph20166566

**Published:** 2023-08-12

**Authors:** Marta I. Mendes, Sara C. Cunha, Iméne Rebai, José O. Fernandes

**Affiliations:** 1LAQV-REQUIMTE, Laboratory of Bromatology and Hydrology, Department of Chemical Sciences, Faculty of Pharmacy, University of Porto, 4050-313 Porto, Portugal; up201809668@ff.up.pt; 2Laboratory of Toxicology, Faculty of Medicine, Salah Boubnider University 3, Constantine 5000, Algeria; imene.rebai@univ-constantine3.dz

**Keywords:** multiple mycotoxin analysis, urine biomonitoring, exposure assessment

## Abstract

Mycotoxins, produced by fungi as secondary metabolites, have the potential to induce both short-term and long-term toxic consequences in animals and humans. The present study aimed to determine multi-mycotoxin levels in Algerian workers using urine as the target. A method based on a QuEChERS (quick, easy, cheap, effective, rugged, and safe) extraction procedure followed by liquid chromatography–tandem mass spectrometry (LC-MS/MS) was optimized and validated for the determination of eleven mycotoxins in 96 urine samples. Different sorbents were tested to be used in the dispersive solid-phase extraction (d-SPE) cleanup step of QuEChERS. The final method was fit-for-purpose and showed good analytical performance in terms of specificity, linearity, and precision. All samples contained at least two mycotoxins, and toxin-2 (T-2) was the most common, being found in 92.7% of the samples, followed by zearalenone (ZEN) in 90.6% of positive samples, and ochratoxin A (OTA) in 86.4%. T-2 levels ranged from 0.3 μg/L to 36.3 μg/L, while OTA ranged from 0.3 μg/L to 3.5 μg/L, and ZEN ranged from 7.6 μg/L to 126.8 μg/L. This was the first mycotoxin biomonitoring study carried out in the Algerian population. The findings highlight the need for accurate data for better risk assessment and for the development of better regulation to manage mycotoxin contamination in this country.

## 1. Introduction

A major challenge in the 21st century is the protection of consumers against risks arising from the consumption of contaminated food. Amongst the food contaminants most worrying for their impact on human health are mycotoxins, a chemically diverse group of toxic secondary fungal metabolites that can be found in a wide array of food commodities, originating from fungal infection and proliferation in the field or during storage. Mycotoxins exhibit in vivo toxicity towards vertebrates after entering via a natural route (i.e., ingestion, inhalation, and dermic contact) and may cause acute and/or chronic severe adverse effects in human health, even at the low levels in which they are usually present in food [1]. Out of the 400 mycotoxins identified so far, only a small fraction is under legal regulation and is subjected to regular monitoring, with aflatoxins, fumonisins, trichothecenes, zearalenone (ZEN), and ochratoxin A (OTA) being the ones most often tested [2,3]. In some countries such as Algeria, however, the number of mycotoxins regulated is even more limited, focusing only on aflatoxins [4].

Nevertheless, some studies have reported mycotoxin occurrence in foods and feeds in Algeria with levels that in some cases were higher than the legal limit established in the European Union (EU) [5,6,7,8]. A study by Madjoubi et al. [5] found that 21 maize samples, 7 wheat samples, and 1 maize sample from Algerian markets presented levels of fumonisins (FB1 + FB2), ZEN, and deoxynivalenol (DON), respectively, above the maximum allowed level established by the EU [2]. Riba et al. [6] found several samples of wheat grains and wheat-derived products exceeding the OTA limits established by the EU (3 μg/kg for raw cereals and 5 μg/kg for cereal products). In a previous study [7], the same group had reported that 90% of nut samples analyzed were contaminated with aflatoxins, with concentrations of AFB1 ranging from 0.2 to 20.52 μg/kg, although with only one sample exceeding the maximum limit allowed by Algerian and EU regulations (10 μg/kg).

Assessing dietary exposure, combining contaminant levels and consumption data, is the process usually applied to estimate human exposure to mycotoxins. However, in the last two decades, the direct human biomonitoring of biological fluids such as urine has been proposed as an alternative approach to assess health risk as it is non-invasive and provides accurate exposure assessment and since it covers exposure from all possible sources [9,10]. This approach has been scarcely applied to African countries, particularly Algeria.

The quantification of mycotoxins in urine requires extraction and a cleanup procedure, followed by a chromatographic step for the separation and quantification of the compounds, usually by means of liquid chromatography with tandem mass spectrometry (LC-MS/MS) [11,12,13]. Protocols need to be appropriate to provide adequate sensitivity, that is, using acceptable limits of detection (LODs) and limits of quantification (LOQs), and be effective, fast, and economic. An optimized sample pre-treatment guides accurate and consistent results. Extraction protocols in urine are mostly based on LLE (liquid–liquid extraction), but recently these protocols have been optimized to allow lower volumes and provide a faster analysis. Some techniques that resulted from this optimization are SALLE (salting-out liquid–liquid extraction), QuEChERS (quick, easy, cheap, effective, rugged, and safe), and DLLME (dispersive liquid–liquid microextraction) [14,15,16]. QuEChERS involves extraction, usually with acetonitrile, in the presence of a substantial quantity of inorganic salts in order to provide the separation of the acetonitrile phase from the aqueous media. The following cleanup step is performed by dispersive solid-phase extraction via suitable sorbents [17,18]. Consequently, QuEChERS has the advantages of being cheap, time-efficient, and simple to operate, while providing good recovery values, and, recently, it has been widely employed in mycotoxin biomonitoring analysis in urine, as reported by Martins et al. [13] and Pallarés et al. [10].

The current study aimed to develop a sensitive and accurate method for the analysis of 11 mycotoxins and metabolites (deoxynivalenol (DON), deoxynivalenol-3-glucoside (DON-3gluc), deepoxy-deoxynivalenol (DOM-1), T-2 toxin (T-2), HT-2 toxin (HT-2), OTA, ZEN, α-zearalenol (α-ZEL), aflatoxin B1 (AFB1), aflatoxin B2 (AFB2), fumonisin B1 (FB1)) in urine using LC-MS/MS. These biomarkers were monitored in the first morning urine samples of Algerian workers from a plastic factory in order to elucidate their exposure to mycotoxins. The workers of this company included adults of different social and economic areas, representing part of the labored society of Algeria. 

## 2. Materials and Methods

### 2.1. Reagents and Standard Solutions

The standards of AFB1, AFB2, T-2, HT-2, OTA, ZEN, and α-ZEL were obtained from Sigma (West Chester, PA, USA) and Fluka (West Chester, PA, USA), all with purity higher than 97%. DON, DON-3-G, DOM-1, and FB1 were also purchased from Sigma, with purity higher than 92.5%. 

The surrogate standard ochratoxin A-(phenyl-d5) (OTAd5) was purchased from Sigma at 10 mg/L, with purity higher than 95%. A working solution was prepared at 500 μg/L in MeOH. A 13C15-DON (deoxynivalenol-13C15) solution, employed as an internal standard, was acquired from Fluka at 25 μg/L in acetonitrile.

Two mixed solutions were prepared in solvent B: a mixture of DON, DOM-1, DON-3-G, and ZEN at 1 mg/L and a mixture of AFB1, AFB2, FB1, OTA, T-2, HT-2, and α-ZEL at 400 μg/L. The standards and solutions were always kept at −18 °C when not in use.

Acetonitrile, formic acid, and methanol were purchased from Merck (Darmstadt, German). Anhydrous magnesium sulfate was purchased from Sigma (West Chester, PA, USA) and treated at 500 °C for 5 h before use. Octadecylsilica (C18, particle size 55–105 mm) was purchased from Waters (Milford, MA, USA). Ultrapure water was obtained by purification with a “Seral” system (Seral Pur, Pro 90 CN, sourced from Seral, Ransbach-Baumbach, Germany) for use in the mobile phase.

### 2.2. Sample Collection

To study mycotoxin exposure, first morning urine samples were collected, between 08 a.m. and 10 a.m., from 96 Algerian adults that work in a plastics factory situated in the industrial zone of Didouche Mourad in Hamma Bouziane. All samples correspond to male individuals, aged from twenty-eight to sixty years old, who worked eight hours a day. The urine samples collected were maintained at −20 °C throughout the transportation process. 

The participants were not subjected to any dietary restrictions prior to sample collection, and their specific diets are unknown. However, it is worth mentioning that, in the region where the study took place, it is common for people to consume homemade bread, nuts, and tea, foods that may contain high amounts of mycotoxins [1,19]. Additionally, the origin and storage conditions of these foods are unknown, although a significant portion of the food reserves are imported from China. All participants answered a questionnaire regarding their demographics (age, body mass index, and place of residence) (Appendix A). The average age of the workers in the study was 43 ± 8.03 years. The average body mass index was 24.72 ± 3.12, amidst a range of 16.01–34.60, with 5.2% being obese (BMI > 30), 42.7% being overweight (BMI 25–30), 50.0% being normal (BMI 18.5–24.9), and 2.1% being underweight (BMI < 18). 

This study’s authorization was formulated by the concerned authorities and followed the standards of the Ethics Committee of the Scientific Committee of the Pharmacy Department of Constantine (Algeria) for Clinical Investigations (Ref CS/CE/01/2019).

### 2.3. LC-MS/MS Conditions

Chromatographic analysis was performed as described by Caldeirão et al. [20]. LC-MS/MS assays were performed using the Waters Alliance 2695 HPLC system (Waters, Milford, MA, USA), which comes with a Quattro Micro triple quadrupole mass spectrometer (Waters, Manchester, UK). For separation, we used an ACQUITY UPLC BEH C18 Column, 130 Å, 1.7 μm, 2.1 mm × 100 mm (Waters, Manchester, UK) maintained at 30 °C.

Mobile phase A was made up of 94% water, 5% methanol, and 1% acetic acid (94:5:1 (*v*/*v*/*v*) and 5 mM ammonium acetate), and mobile phase B was made up of 97% methanol, 2% water, and 1% acetic acid (97:2:1 (*v*/*v*/*v*)). A gradient elution was performed using these two mobile phases, starting at 95% mobile phase A with a linear decrease to 35% in 7 min. In the following 4 min, mobile phase A was decreased to 25%, and, at 11 min, an isocratic gradient of 100% of mobile phase B started for 2 min. Initial column conditions were reached at 25 min and maintained for 2 min until the next injection. The flow rate was adjusted to 0.3 mL/min.

The MS/MS acquisition was performed in positive-ion mode with multiple reaction monitoring (MRM). Argon 99.995% (Gasin, Porto, Portugal), with a pressure of 2.9 × 10^−3^ mbar in the collision cell, was used as the collision gas. Capillary voltages of 3 kV were used in the positive ionization mode. Nitrogen was the desolvation gas and cone gas, with flows of 350 and 60 L/h, respectively. The desolvation temperature was set to 350 °C, and the source temperature was set to 150 °C. Data were collected using MassLynx 4.1.

The precursor and product ions were selected according to different conjugations of cone voltages and collision energies to obtain the most advantageous MRM transition for accurate mycotoxin identification. The optimized LC-MS/MS parameters for each mycotoxin analyzed are listed in Appendix A.

### 2.4. Extraction and Cleanup

Mycotoxins were extracted using the previously developed QueChERS method [21], with some modifications. Before analysis, the urine samples were thawed at room temperature, and 120 μL of OTAd5 (500 μg/L) was added. The extraction was performed with a 3 mL mixture of 99% acetonitrile and 1% formic acid (*v*/*v*). A total of 1 g of MgSO4 and 0.25 g of sodium acetate were also added. The resulting mixture was vortexed and then agitated for 15 min (rotary shaker Multi RS- 60 Biosan, sourced from Biosan, Riga, Latvia) and centrifuged at 4000× *g* for 5 min. Then, 2 mL of organic phase was transferred to a tube already containing 200 mg of C18 and centrifuged, and 1.3 mL of the supernatant was aspirated and evaporated at 45 °C under a stream of nitrogen. The resulting residue was resuspended with 500 μL of mobile phase B, along with 12.5 μL of 13C15-deoxynivalenol (13DON15) at 25 μg/L, and it was finally injected into the LC-MS/MS system.

### 2.5. Method Validation

Matrix-matching calibration curves were generated for all mycotoxins, using six different concentration levels. Based on previous works [11,14,22], two distinct levels of concentration ranges were evaluated: from 6.75 to 225 μg/L for DON, DON-3-G, DOM-1, and ZEN and from 0.25 to 5.0 μg/L for AFB1, AFB2, FB1, T-2, HT-2, OTA, and α-ZEL. The LOQ was calculated as the concentration of analyte providing a signal-to-noise of 10, and the LOD was calculated as the concentration of analyte, providing a signal-to-noise of 3. The intra-day precision of each mycotoxin was measured on the same day in five replicate experiments at two different levels of concentration. For DON, DON-3-G, DOM-1, and ZEN, the first level was at 50 μg/L, and the second level was at 100 μg/L. For AFB1, AFB2, FB1, T-2, HT-2, OTA, and α- ZEL, the first level was at 2 μg/L, and the second level was at 4 μg/L. 

### 2.6. Statistical Analysis

To determine whether the mycotoxin data followed a parametric or non-parametric distribution, a Kolmogorov–Smirnov test was performed. The results revealed a non-normal distribution, prompting the selection of a nonparametric Mann–Whitney U-test to compare the mycotoxin concentration between rural and urban areas given the sample size and distribution. Fisher’s exact test was applied to compare mycotoxin frequency between both groups. Statistical significance was determined by setting a significance level of 0.05. All statistical analyses were conducted using the SPSS statistical package, version 27.0 (IBM Corporation, New York, NY, USA).

## 3. Results and Discussion

### 3.1. Cleanup Optimization

Two different solid sorbents, EMR-lipid and C18, were assessed for the dispersive cleanup step. Thus, 2 mL of extract obtained from the QuEchERs extraction of 1.5 mL of spiked urine (25 μg/L) with 3 mL of acidified acetonitrile (1% formic acid) was treated with 200 mg of EMR-lipid or 200 mg of C18. After this, the extracts were centrifuged, and 1 mL of supernatant was evaporated at 45 °C under a stream of nitrogen. Once evaporated, the residues were resuspended with 500 μL of mobile phase B so they could be injected into the LC-MS/MS system. The results obtained were compared in terms of analytical signal (see Appendix A). 

Despite the slightly better results obtained with EMR-lipid, this sorbent requires an activation step and is more expensive than C18. Consequently, without compromising the results, C18 was chosen as the sorbent.

### 3.2. Analytical Validation

The performance results are listed in Table 1. All analytes showed good linear responses, with r values above 0.962 for all mycotoxins, except HT-2, for which r = 0.907 (Table 1).

The RSD (relative standard deviation) values obtained were satisfactory at both levels of concentration, being lower than 20% in most cases, with the exception of AFB1, which presented a higher RSD%. 

The LOQ and LOD ranged, respectively, from 6.75 μg/L and 2.05 μg/L for DON, DON-3-G, DOM-1, and ZEN to 0.25 μg/L and 0.08 μg/L for AFB1, AFB2, FB1, T-2, HT-2, OTA, and α-ZEL. Our LOQ and LOD were within the range reported in the literature [11,23,24] but slightly higher than those reported in Martins et al. [13], Martins et al. [25], and Huybrechts et al. [26].

### 3.3. Sample Results

#### 3.3.1. Levels of the Mycotoxins and Their Metabolites

The frequency of positive samples is shown in Table 2, along with the average, minimum, and maximum concentrations of each mycotoxin. 

In this study, DON was not found in any samples (Table 2), having lower prevalence than previously reported in the literature (Table 3). In a study by Vidal et al. [27], it was observed that DON is rapidly excreted, and therefore, to obtain a more representative analysis of DON exposure, at least 16 h of urine collection is suggested. To further support this, a Portuguese study by Martins et al. [13] analyzed both first morning urine and 24 h urine samples and found that DON occurrence was significantly higher in the 24 h urine samples. Nevertheless, DON metabolites, DON-3-G and DOM-1, were found, in this study, in 45.8% and 76.0% of samples, respectively, with average levels of 13.28 μg/L and 47.97 μg/L, ranging from 6.8 to 37.8 μg/L and from 6.9 to 189.1 μg/L. These levels are comparable to those found in Rwanda [28] and Spain [29] and higher than those found in Portugal [13]. 

T-2 was found in 89 out of 96 samples (92.7%), with levels ranging from 0.3 to 36.3 μg/L (average of 8.37 μg/L). Its metabolite, HT-2, also presented a high prevalence (77.1%) and an average concentration of 2.05 μg/L, ranging from 0.3 to 11.0 μg/L. These compounds, in general, were not reported in urine biomonitoring studies [13,30,31].

ZEN was found in 89.6% of the samples, with an average concentration of 28.87 μg/L, ranging from 7.6 to 126.8 μg/L. The average level in the present study was higher than in previous studies in Rwanda (average of 1.58 μg/L) [28], South Africa (0.20 μg/L) [32], Portugal (1.30 μg/L) [13], and Spain (6.70 μg/L) [29]. Regarding the α-ZEL metabolite, found in 24.9% of samples, the average was 0.43 μg/L (from 0.3 to 1.0 μg/L). This level was higher than that obtained in South Africa (0.25 μg/L) [32] but lower than that obtained in Chile (41.80 μg/L) [24] and in Portugal (2.70 μg/L) [13].

OTA was detected in 86.4% of samples, with levels between 0.3 μg/L and 3.5 μg/L and an average of 0.82 μg/L, which is higher than the average reported in a study in Portugal [13] and the average levels reported in African countries, as indicated in Table 3 [28,30,32,33,34], but lower than the average reported in Chile (1.30 μg/L) [24].

Regarding aflatoxins, 18 samples (18.8%) were positive for AFB1, with an average concentration of 0.82 μg/L (from 0.3 to 4.7 μg/L), and 10 samples (10.4%) were positive for AFB2, with an average of 1.17 μg/L (from 0.3 to 5.8 μg/L). These results show a higher prevalence and concentration for aflatoxins in Algeria in comparison to other countries [24,25,28]. 

FB1, present in 36.5% of samples, ranged from 0.5 to 96.2 μg/L, with an average of 12.99 μg/L, similar to the average reported in Ivory Coast (15.30 μg/L) [34].

**Table 3 ijerph-20-06566-t003:** Comparison of the results obtained with other studies that use mycotoxin biomarkers in urine.

	Mycotoxin	Algeria	Chile [24]	Ivory Coast [34]	Nigeria [30]	Portugal [13,25]	Rwanda [28]	South Africa [32]	Spain [29]
Mycotoxin Prevalence (%)	DON	0	55	21	0.8	30	19	87	23
	DON-3-G	46	-	-	5	24	48	-	-
	DOM-1	76	-	0	-	32	24	-	53
	ZEN	91	1	37	0.8	57	30	100	40
	α-ZEL	25	8	-	-	5	-	92	43
	OTA	86	1	-	28	27	71	96	3
	T-2	93	-	-	-	ND	-	-	-
	HT-2	77	-	-	-	ND	-	-	-
	AFB1	19	8	-	-	2	8	-	-
	AFB2	10	-	-	-	0	-	-	-
	FB1	37	-	27	13.3	-	30	-	-
Mycotoxin Average (µg/L)	DON	-	60.70	10.00	2.00	0.38	18.80	4.94	9.07
	DON-3-G	13.28	-	-	3.50	0.25	5.88	-	-
	DOM-1	47.97	-	-	-	0.23	35.00	-	20.28
	ZEN	28.59	1.10	-	0.30	1.30	1.58	0.20	6.70
	α-ZEL	0.43	41.80	-	-	2.70	-	0.25	27.44
	OTA	0.82	1.30	0.42	0.20	0.01	0.03	0.02	11.73
	T-2	8.37	-	-	-	ND	-	-	-
	HT-2	2.05	-	-	-	ND	-	-	-
	AFB1	0.82	0.30	-	-	0.003	0.01	-	-
	AFB2	1.17	-	-	-	<LOQ	-	-	-
	FB1	12.99	-	15.30	4.60	0.24	0.01	-	-

DON, Deoxynivalenol; DON-3-G, deoxynivalenol-3-glucoside; DOM-1, deepoxy-deoxynivalenol; ZEN, zearalenone; α-ZEL, α-zearalenol; OTA, ochratoxin A; T-2, T-2 toxin; HT-2, HT-2 toxin; AFB1, aflatoxin B1; AFB2, aflatoxin B2; FB1, fumonisin B1; ND, not detected.

#### 3.3.2. Co-Occurrence

Food contamination by multiple mycotoxins is very common as certain fungal species have the ability to produce various types of mycotoxins simultaneously and because food can also be contaminated by multiple fungal species. This is a serious issue for public health since current legislation does not account for the hazards of multi-mycotoxin exposure and because mycotoxins can have additive or synergistic effects, so their toxicity does not always correspond to individual toxicities summed together. Therefore, when evaluating mycotoxin exposure, there is a necessity to consider the co-occurrence of mycotoxins.

Several studies conducted in the Mediterranean region, where Algeria is located, revealed a high prevalence of co-occurring mycotoxins in cereals. In Morrocco, 51% of tested samples were found to be co-contaminated with two to six mycotoxins [35]. Similarly, in Spain [36] and in Italy [37], 65% and 81% of samples under study, respectively, were contaminated by a minimum of two mycotoxins. As for Algeria, a study by Mahdjoubi et al. [5] found that 50% of samples were contaminated with two to nine mycotoxins. Given the potential health risks associated with the consumption of co-contaminated cereals, further research is needed to develop effective strategies to mitigate these risks, especially in this area. 

In order to analyze the prevailing mycotoxin patterns within the samples, an upset plot was generated (Figure 1). The results reveal a diverse array of mycotoxin co-occurrence, with the most dominant combination being DOM-1 + HT-2 + OTA + ZEN + T-2, which was detected in nine samples (9.4% of samples). Notably, these five mycotoxins were simultaneously present in a total of 43 samples, representing 44.8% of samples. These findings indicate a significant co-occurrence of mycotoxins in the analyzed samples, with every sample containing at least two positive mycotoxins. On average, each sample contained 5.6 mycotoxins, with the highest count observed in sample number seventy-six, with a total of 9 positive mycotoxins (DON-3-G + DOM-1 + ZEN + α-ZEL + T-2 + HT-2 + OTA + AFB1 + AFB2).

#### 3.3.3. Distribution of Mycotoxins and Their Metabolites

In this study, it was possible to distinguish samples from rural areas from those of urban areas, as presented in Table 4. There are thirteen samples from subjects from rural areas and 82 samples from subjects from urban areas. There is one sample that is not specified in this regard (sample n° 48). 

It was expected that people from rural areas would have a higher occurrence of some mycotoxins, especially DON, T-2, and ZEN, which are in grains and typically more consumed in these areas. In fact, DOM-1 had a higher frequency and a higher average concentration in samples from rural areas (*p* < 0.05): all these samples tested positive, with an average of 88.58 μg/L, while, in urban areas, 43.9% of samples tested positive, with an average of 39.31 μg/L. In contrast, T-2 had a higher frequency and a higher concentration in the urban group (*p* < 0.05), for which 98.8% of samples tested positive, with an average of 9.01 μg/L, whilst, in the rural group, 69.2% of samples tested positive, with an average of 1.37 μg/L. HT-2, a metabolite of T-2, also seemed to be more frequent in urban samples (81.7% in urban samples vs. 38.5% in rural samples; *p* < 0.05), but average concentrations were similar between both groups (2.08 μg/L in urban and 2.06 μg/L in rural). ZEN also had a similar average concentration in both groups (29.97 μg/L in urban and 20.32 μg/L in rural), but a higher frequency in the urban group (93.9% positive samples vs. only 61.5% in the rural group; *p* < 0.05). As for α-ZEL, it was found more frequently in the rural group (76.9% in rural vs. 15.85% in urban; *p* < 0.05), but the average concentrations in each group were similar (0.38 μg/L in rural and 0.48 μg/L in urban). OTA (100% positive with an average of 0.92 μg/L in rural vs. 84.1% positive with an average of 0.82 μg/L in urban) and DON-3-G (53.8% positive with an average of 11.71 μg/L in rural vs. 43.9% positive with an average of 13.66 μg/L in urban) had similar percentages of positive samples and similar average concentrations in both groups (*p* > 0.05). Regarding FB1, the differences in mycotoxin frequency and concentration between both groups were not statistically significant (1.37 μg/L in rural samples, *p* > 0.05). AFB1 was only found in the urban group (15.4% of urban samples, with an average of 0.85 μg/L), However, this does not correlate with urban areas having a higher incidence of this mycotoxin as it was found in significantly more samples from urban areas than in those from rural areas. AFB2 was present in two rural samples (15.4% with an average of 0.85 μg/L) and eight urban samples (9.8% with an average of 1.25 μg/L).

The findings of this study showed that some mycotoxins, such as T-2 and ZEN, had higher incidences in samples from urban areas, thereby challenging the common belief that rural areas are more exposed to mycotoxins. This may be due to several factors, including the longer times of grain storage in silos in urban agriculture, the higher levels of pollution that may contribute to a higher incidence of fungal infections in crops, or the fact that urban buildings are more likely to have high levels of mold [38]. However, it is important to highlight that there are significant differences in the number of samples collected from each group, which makes it a challenge to draw definitive conclusions about which location is more susceptible to mycotoxin exposure in this environment. Furthermore, it is important to exercise caution when generalizing mycotoxin incidence in different locations as it can vary greatly depending on various complex factors that may fluctuate in specific situations, so it is essential to consider the unique environmental and socioeconomic circumstances of each area when examining mycotoxin exposure. 

## 4. Conclusions

To the best of our knowledge, this was the first study to evaluate mycotoxin exposure in the Algerian population. The results are worrying, showing high mycotoxin exposure across workers coming from different socioeconomic backgrounds, emphasizing the need for awareness of this issue and preventive measures. In Algeria, where a significant number of cereals are imported, and where long shipping trips have been known to increase the possibility of fungal growth, there is an increased need to control mycotoxin-producing fungi and to monitor storage and harvesting conditions. It is also urgent to implement limits for mycotoxins in food as, currently, only legislation regarding aflatoxins in cattle feed, nuts, and cereals exists. This study highlights the importance of addressing mycotoxin exposure in Algeria and serves as a call for action for Algerian authorities to implement measures to reduce exposure and protect public health.

## Figures and Tables

**Figure 1 ijerph-20-06566-f001:**
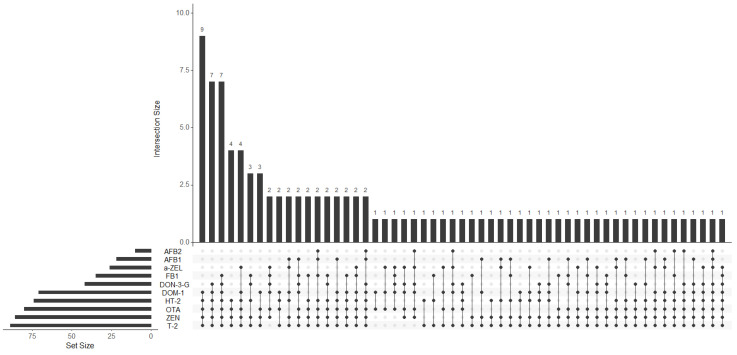
Upset Plot showcasing mycotoxin co-occurrence. Positive mycotoxins are represented by balck dots and the presence of multiple mycotoxins is indicated by connected bars. DON-3-G, deoxynivalenol-3-glucoside; DOM-1, deepoxy-deoxynivalenol; ZEN, zearalenone; α-ZEL, α-zearalenol; OTA, ochratoxin A; T-2, T-2 toxin; HT-2, HT-2 toxin; AFB_1_, aflatoxin B1; AFB2, aflatoxin B2; FB1, fumonisin B1.

**Table 1 ijerph-20-06566-t001:** Calibration range, correlation coefficient (r), LOQ, LOD, RSD (%), and extraction yield (%) of the mycotoxins.

Mycotoxins	Calibration Range (μg/L)	Correlation Coefficient (r)	LOQ (μg/L)	LOD (μg/L)	Repeatability (RSD %)	
					First Level	Second Level
DON	6.75–225	0.997	6	2.05	3.1 *	1.1 *
DON-3-G	6.75–225	0.981	6	2.05	1.1 *	0.2 *
DOM-1	6.75–225	0.997	6	2.05	6.4 *	0.9 *
ZEN	6.75–225	0.997	6	2.05	7.5 *	26.8 *
α-ZEL	0.25–5	0.978	0.2	0.08	6.2 **	0.8 **
OTA	0.25–5	0.999	0.2	0.08	22.3 **	3.1 **
T-2	0.25–5	0.997	0.2	0.08	1.9 **	2.8 **
HT-2	0.25–5	0.907	0.2	0.08	17.9 **	14.1 **
AFB1	0.25–5	0.997	0.2	0.08	43.0 **	33.3 **
AFB2	0.25–5	0.995	0.2	0.08	6.0 **	6.8 **
FB1	0.25–5	0.962	0.2	0.08	0.01 **	0.02 **

LOD, Limit of detection; LOQ, limit of quantification; RSD, relative standard deviation; DON, deoxynivalenol; DON-3-G, deoxynivalenol-3-glucoside; DOM-1, deepoxy-deoxynivalenol; ZEN, zearalenone; α-ZEL, α-zearalenol; OTA, ochratoxin A; T-2, T-2 toxin; HT-2, HT-2 toxin; AFB1, aflatoxin B1; AFB2, aflatoxin B2; FB1, fumonisin B1. * First level at 50 μg/L and second at 100 μg/L; ** first level at 2 μg/L and second at 4 μg/L.

**Table 2 ijerph-20-06566-t002:** Frequency of positive samples (%) and average, minimum, and maximum concentrations (RSD, relative standard deviations).

Mycotoxin	Frequency (%)	Average ± RSD (µg/L)	Min (µg/L)	Max (µg/L)
DON	0	ND	ND	ND
DON-3-G	44 (45.8)	13.28 ± 6.5	6.8	37.80
DOM-1	73 (76.0)	47.97 ± 18.8	6.9	189.1
ZEN	86 (89.6)	28.87 ± 20.9	7.6	126.8
α-ZEL	23 (24.9)	0.43 ± 0.2	0.3	1.0
OTA	83 (86.4)	0.82 ± 0.5	0.3	3.5
T-2	89 (92.7)	8.37 ± 7.6	0.3	36.3
HT-2	74 (77.1)	2.05 ± 1.9	0.3	11.0
AFB1	18 (18.8)	0.82 ± 0.9	0.3	4.7
AFB2	10 (10.4)	1.17 ± 1.5	0.3	5.8
FB1	35 (36.5)	12.99 ± 17.1	0.5	96.2

DON, Deoxynivalenol; DON-3-G, deoxynivalenol-3-glucoside; DOM-1, deepoxy-deoxynivalenol; ZEN, zearalenone; α-ZEL, α-zearalenol; OTA, ochratoxin A; T-2, T-2 toxin; HT-2, HT-2 toxin; AFB1, aflatoxin B1; AFB2, aflatoxin B2; FB1, fumonisin B1; ND, not detected.

**Table 4 ijerph-20-06566-t004:** Differences in occurrence between rural and urban areas.

	Rural (13)		Urban (82)		*p*-Value	
	Positive Samples (%)	Average (µg/L)	Positive Samples (%)	Average (µg/L)	Frequency ^a^	Concentration ^b^
DON-3-G	7 (53.8)	11.71	36 (43.9)	13.66	0.559	0.834
DOM-1	13 (100)	88.58	59 (71.9)	39.31	0.034	<0.01
ZEN	8 (61.5)	20.32	77 (93.9)	29.97	0.004	0.155
α-ZEL	10 (76.9)	0.38	13 (15.85)	0.48	<0.01	0.148
OTA	13 (100)	0.92	69 (84.1)	0.82	0.203	0.793
T-2	9 (69.2)	1.37	81 (98.8)	9.01	<0.01	<0.01
HT-2	5 (38.5)	2.06	67 (81.7)	2.08	0.01	0.848
AFB1	0	<LOQ	18 (22.0)	0.82	0.12	-
AFB2	2 (15.4)	0.85	8 (9.8)	1.25	0.623	0.533
FB1	3 (23.1)	1.37	33 (40.2)	13.67	0.123	0.067

LOQ, Limit of quantification; DON, deoxynivalenol; DON-3-G, deoxynivalenol-3-glucoside; DOM-1, deepoxy-deoxynivalenol; ZEN, zearalenone; α-ZEL, α-zearalenol; OTA, ochratoxin A; T-2, T-2 toxin; HT-2, HT-2 toxin; AFB1, aflatoxin B1; AFB2, aflatoxin B2; FB1, fumonisin B1. ^a^ Fisher’s exact test; ^b^ Mann–Whitney U-test.

## Data Availability

The data that support the findings of this study are available in the Appendix A of this article.

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
