# Peer review of "Algerian Workers’ Exposure to Mycotoxins—A Biomonitoring Study"

_ijerph, 2023, doi:10.3390/ijerph20166566_

Round 1

Reviewer 1 Report

The manuscript developed an method for determination of 11 mycotoxins in urine based on a QuEChERS and LC-MS/MS and applied to urine of Algerian worker..the work is new and of interest to the readership of Int. J. Environ. Res. Public Health. A significant amount of data is presented, and the conclusions are supported by these data. However, the data presentation and interpretation must be improved, and some technical details must be clarified prior to further consideration for publication.I recommend major revision. 
1)LC-MS/MS (liquid-chromatography tandem mass spectrometry)  change to liquid-chromatography tandem mass pectrometry(LC-MS/MS);

DON-3-gluc change to DON-3G.

2) The average body mass of worker have been included

3) these data on of added recovery of LOQ, 10LOQ should been added to analytical validation

4) Is the concentration in table 2 a single test data? The sample should be analyzed more than twice, the The result should have a relative deviation.

Is “(%)” Represents frequency? It should be explained in the table annotations

5) Suggest using a table to represent the results of levels of mycotoxin

 between this manuscript and other literature.

6)The possible reasons for the differences between rural and urban areas should be analyzed and introduced.

NO

Author Response

Dear Editor,

We appreciate your comments and those of the reviewers on our manuscript. All of these comments were very helpful for revising and improving our paper, which has been revised in accordance with the recommendations. All the changes in the revised manuscript are highlighted and a detailed point-by-point response to each comment raised in the review process is provided below.

Thank you very much for your attention and please feel free to contact us with any further questions.

Reviwer 1

The manuscript developed an method for determination of 11 mycotoxins in urine based on a QuEChERS and LC-MS/MS and applied to urine of Algerian worker..the work is new and of interest to the readership of Int. J. Environ. Res. Public Health. A significant amount of data is presented, and the conclusions are supported by these data. However, the data presentation and interpretation must be improved, and some technical details must be clarified prior to further consideration for publication.I recommend major revision. 

1)LC-MS/MS (liquid-chromatography tandem mass spectrometry)  change to liquid-chromatography tandem mass pectrometry(LC-MS/MS);

The changed (line 20) was done as suggested.

DON-3-gluc change to DON-3G.

The modification was made across the manuscript, tables, figures and supplementary material.

2) The average body mass of worker have been included

The average body mass index (BMI) was included (line 130). We don’t have information about body mass, only BMI. (BMI of each sample is in Table S1 (supplementary materials).

3) these data on of added recovery of LOQ, 10LOQ should been added to analytical validation

The correction was made as suggested.

4) Is the concentration in table 2 a single test data? The sample should be analyzed more than twice, the The result should have a relative deviation.

The relative deviation was added as suggested.

Is “(%)” Represents frequency? It should be explained in the table annotations

The explanation was included in the table (line 270).

5) Suggest using a table to represent the results of levels of mycotoxin between this manuscript and other literature.

А table is presented in line 272.

6)The possible reasons for the differences between rural and urban areas should be analyzed and introduced.

The possible reasons are now listed in the lines 365-367 and 390-403.

Reviewer 2 Report

The study examined the different mycotoxins in the urine of workers of an Algerian plastic factory.  It’s very beneficial to obtain exposure information on mycotoxin for policymaking.  Although it is the first evaluation of mycotoxin exposure in Algeria, I have some comments.

The title is too general, need more specific.

The demographic table was missing.

The results part does not have the description of the corresponding findings. In contrast, the discussion part is mainly focused on the results. More editing work is needed.

Figure 1 is not informative and hard to compare. I suggest using upset plot which can show the cooccurrence of multiple mycotoxins.

Plastic industry is not a good term. Line 42 “restrict“ is not proper. The “contaminate” was used in samples which is misleading. 

Author Response

Reviwer2

The study examined the different mycotoxins in the urine of workers of an Algerian plastic factory.  It’s very beneficial to obtain exposure information on mycotoxin for policymaking.  Although it is the first evaluation of mycotoxin exposure in Algeria, I have some comments.

The title is too general, need more specific.

Title was change as suggested.

The demographic table was missing.

The demographic table is in supplementary materials (Table S1).

The results part does not have the description of the corresponding findings. In contrast, the discussion part is mainly focused on the results. More editing work is needed.

 For better understanding, results and discussion were joined.

Figure 1 is not informative and hard to compare. I suggest using upset plot which can show the cooccurrence of multiple mycotoxins.

 Upset plot was created and is in line 332, and is described in lines 323-331.

Plastic industry is not a good term. Line 42 “restrict“ is not proper. The “contaminate” was used in samples which is misleading. 

The term Plastic industry was changed to plastic factory. “Restrict” at line 47 was changed to “limited”.  Contaminate was deleted.

Reviewer 3 Report

This manuscript has the value of academic and public health management, and the citations of the literature are rich and complete. However, it is recommended to modify or supplement the following information:

This manuscript is not a review paper, and its title: “Incidence of mycotoxin in Algerian workers” is too general to correspond with the abstract and content. Recommendations should focus more on analytical methods, results and key findings.

Figure 1- Occurrence of mycotoxins in each sample. This figure is very unclear, please provide a clearer graphic to understand its content. In addition, please classify and describe samples 1-95.

Haven't received Supplementary Materials related information, please provide.

Author Response

This manuscript has the value of academic and public health management, and the citations of the literature are rich and complete. However, it is recommended to modify or supplement the following information:

This manuscript is not a review paper, and its title: “Incidence of mycotoxin in Algerian workers” is too general to correspond with the abstract and content. Recommendations should focus more on analytical methods, results and key findings.

The title was changed as suggested.

Figure 1- Occurrence of mycotoxins in each sample. This figure is very unclear, please provide a clearer graphic to understand its content. In addition, please classify and describe samples 1-95.

The figure was changed and it’s in line 332, with descriptive text regarding co-occurrence in lines 323-331. Sample characterization is made in Table S1 of Supplementary Materials.

Haven't received Supplementary Materials related information, please provide.

Sorry for that. We hope you have access now

Round 2

Reviewer 2 Report

Thanks for all the responses! All comments have been addressed.

Author Response

Thank you !

Reviewer 3 Report

Figure 1 is still unclear and unreadable. It is recommended to rotate the original figure of "Figure 1- Occurrence of mycotoxins in each sample." in the v1 version by 90 degrees. That is, take the sample number as the vertical axis and the mycotoxin concentration as the horizontal axis, so that it will be clearer.

Author Response

The figure 1 was changed in order to be legible. The change of sample number to the  vertical axis and the mycotoxin concentration to the horizontal axis was not very comprehensible as you can see in the attached, due to the huge number of samples, but if need to change the figure again please let us know. 
